# Polymeric Coatings for Magnesium Alloys for Biodegradable Implant Application: A Review

**DOI:** 10.3390/ma16134700

**Published:** 2023-06-29

**Authors:** G. Keerthiga, M. J. N. V. Prasad, Dandapani Vijayshankar, R. K. Singh Raman

**Affiliations:** 1IITB-Monash Research Academy, Mumbai 400076, Maharashtra, India; keerthiga.ganesan@monash.edu; 2Microstructural Engineering and Mechanical Performance Laboratory, Department of Metallurgical Engineering and Materials Science, Indian Institute of Technology Bombay, Mumbai 400076, Maharashtra, India; mjnvprasad@iitb.ac.in; 3Electrochemistry at Interface Lab, Department of Metallurgical Engineering and Materials Science, Indian Institute of Technology Bombay, Mumbai 400076, Maharashtra, India; v.dandapani@iitb.ac.in; 4Department of Chemical and Biological Engineering, Monash University, Clayton, VIC 3800, Australia; 5Department of Mechanical and Aerospace Engineering, Monash University, Clayton, VIC 3800, Australia

**Keywords:** biodegradable implants, magnesium alloys, polymeric coatings

## Abstract

Magnesium (Mg) alloys are a very attractive material of construction for biodegradable temporary implants. However, Mg alloys suffer unacceptably rapid corrosion rates in aqueous environments, including physiological fluid, that may cause premature mechanical failure of the implant. This necessitates a biodegradable surface barrier coating that should delay the corrosion of the implant until the fractured/damaged bone has healed. This review takes a brief account of the merits and demerits of various existing coating methodologies for the mitigation of Mg alloy corrosion. Since among the different coating approaches investigated, no single coating recipe seems to address the degradation control and functionality entirely, this review argues the need for polymer-based and biodegradable composite coatings.

## 1. Introduction

Metals are the preferred biomaterials among polymers, ceramics, and composites for load-bearing applications as they have superior mechanical properties. The global increase in bone-related trauma, musculoskeletal disorders, bone defects, and clinical treatments has been challenging, particularly for temporary implant applications. Temporary implants provide necessary support until the fractured/damaged bone is restored. When temporary implants (e.g., plates, pins, and screws) are fabricated out of conventional metallic implant materials, e.g., stainless steels (SS), cobalt-chromium (Co-Cr) alloys and titanium (Ti) alloys, they cause stress shielding (because of their higher elastic moduli than natural bone) which is a shortcoming in the use of these alloys as temporary bone implants [1]. Furthermore, when such alloys are used to construct temporary implants, secondary implant removal surgery is required after the implant completes its function [2]. In this context, it would be hugely attractive if temporary implants could be constructed out of such metallic materials that could dissolve away in the physiological environment without generating any dissolution product(s) that could be toxic to human physiology. Alloys of magnesium (Mg), zinc (Zn) and iron (Fe), having no toxic constituents (such as Al), possess such characteristics. Mg (and its alloys) have an advantage due to their mechanical properties being closer to the natural human cortical bone (Table 1), as a result of which the stress shielding effect may be ameliorated. On the other hand, when constructed out of stiffer materials, the implant bears most of load/stress during healing. Thus, the stiffer implant material shields the healing bone from growing with the ability to bear the stress, i.e., stress shielding.

## 2. Need for Coatings for Mg and Its Alloys

Despite the suitable mechanical properties of Mg and its alloys for temporary implant application, their undesirable rapid dissolution/corrosion rate is a concern, and its mitigation is essential for the successful use of Mg alloys as bioimplants. It is necessary to understand Mg’s corrosion behaviour to develop an effective strategy to mitigate its corrosion, such as by application of coatings. When exposed to aqueous solutions, Mg, having very high negative potential and poorly passivating surface, undergoes rapid electrochemical corrosion following the reaction (Equation (1)):(1)$$\mathrm{Mg}+2H_{2}O\;\to \;\mathrm{Mg}{\left(\mathrm{OH}\right)}_{2}+{\;H}_{2}\;(\mathrm{Overall}\;\mathrm{reaction})\;$$

Mg is a highly electronegative metal with a standard electrode potential of −2.37 V (vs. SHE) [3]. As the potential (E)-pH diagram (Figure 1a) shows, Mg-base systems tend to corrode even without the participation of O_2_ in the cathodic process, facilitating water reduction and hydrogen evolution (Figure 1b). Mg undergoes anodic dissolution to form Mg^2+^ ions, while the cathodic half-cell reaction is that of H_2_ evolution, forming a layer of magnesium hydroxide (Mg(OH)_2_). Notably, an excessive rate of H_2_ generation can be dangerous, as it can develop localised gas pockets in poorly aerated bone structures. Though the Mg(OH)_2_ layer is protective [4] (Pilling Bedworth ratio for Mg and its oxide is 0.81, i.e., <1, which means less compact, porous surface metal oxide layer), excessive OH^-^ ions can cause localised alkalinization [5] when pH exceeds 10. Apart from the conventional mechanisms for the corrosion of metals, Mg exclusively undergoes the phenomenon of a negative difference effect (NDE) [6], which is characterised by the increased rate of H_2_ evolution with increasing anodic polarisation (which is in contrast to what would generally be expected in the case of the electrochemical behaviour of most metals [7]). Yet another challenge caused by Mg corrosion in an aqueous environment is the hydrogen embrittlement (HE) and stress corrosion cracking (SCC) that can vastly compromise the toughness of metals and alloys [8,9]. However, NDE, HE and SCC of Mg-alloy bioimplants are outside the scope of this review.

Because bare Mg and its alloys will corrode at unacceptably rapid rates in human body fluid, it becomes essential to improve their in-vivo corrosion resistance [12,13] to retain their mechanical integrity during their use as temporary implants. The rate of degradation also greatly influences cell attachment and growth. Figure 2 shows the schematic of a bone’s required mechanical strength retention during healing when an Mg alloy is used as a corrective/fixture implant material. Until stable hard callus formation is achieved in a fractured bone, Mg implants must retain their mechanical integrity to facilitate healing. However, as seen in Figure 2, Mg will lose its mechanical strength and integrity before the hard callus formation begins. Therefore, there is a crucial need to delay the degradation in the mechanical properties due to in-vivo corrosion by human body fluid, and hence, the need for a temporary barrier until healing completes, such as by application of a suitable coating. The suitable barrier coating can ameliorate Mg’s unacceptably high corrosion rate due to its high electronegativity. 

Inorganic and polymeric materials have been commonly employed as corrosion barrier coatings for Mg alloys. While this review takes a quick stock of the inorganic coatings, it focuses on the polymeric coatings for the temporary bioimplants. Also, for the particular application of temporary bioimplants, the most crucial criteria are biocompatibility, non-toxicity, and biodegradability.

## 3. Biodegradable Surface Coatings for Mg Alloys

For effective use of coated Mg-alloy as temporary implants that dissolve away after the healing of the fractured/damaged bone, both the alloy and the coating need to be biodegradable in the human physiological environment. When a biomaterial comes in immediate contact with the physiological fluid, its surface’s physical and chemical properties dictate the surface bioactivity. As Mg is highly electrochemically active, it undergoes spontaneous dissolution, and the quasi-protective nature of the passive layer that develops on the surface facilitates further localised dissolution and alkalization. Attempts have been made to improve the protective nature of the passive layer by modulating thickness through plasma electrolytic oxidation (PEO), anodization, hydrothermal coating etc., as described in Table 2. Conversion coatings incorporating phosphate onto the hydroxyl group bearing Mg substrate can improve bioactivity.

Various major coating types (metallic, organic, and inorganic) can be developed for corrosion mitigation of Mg alloys (Figure 3 and Figure 4), employing a suite of techniques described in Table 2. Appropriate coating methodology is selected based on the coating material and the merits and demerits of the techniques, as elaborated in Table 2. However, there are critical limitations on the coating chemistry because of the toxicity/biosafety concerns [15,16,17]; for example, Al coating is prohibited since it can cause Alzheimer’s and dementia, even though Al is known to improve corrosion resistance. Besides, any local disruptions in the metallic coating will cause severe galvanic corrosion to the underneath Mg alloy because of Mg’s highly electronegative/anodic nature [18]. On the other hand, inorganic coatings such as diamond-like carbon (DLC) possess high chemical/thermodynamic stability. Hence, they will not fulfil the criterion of eventual dissolution in physiological fluid [19].

**Figure 3 materials-16-04700-f003:**
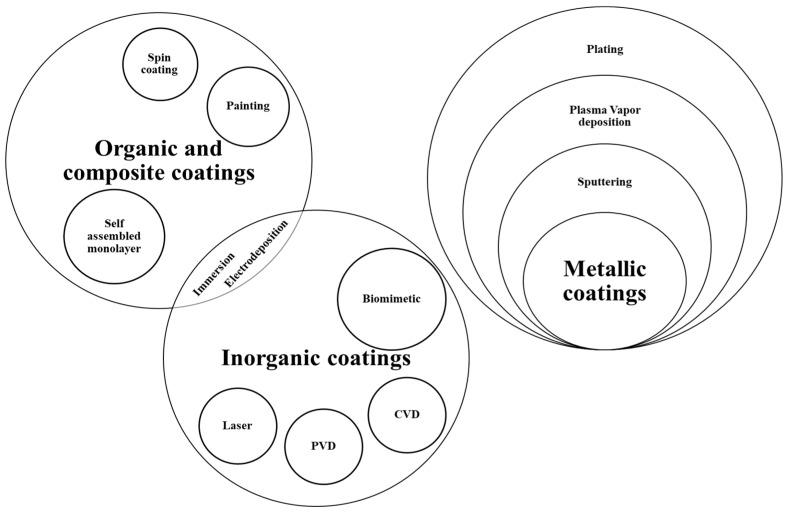
Different coating methodologies to achieve organic, inorganic, metallic and composite coatings on Mg alloys (Redrawn from [20]).

**Figure 4 materials-16-04700-f004:**
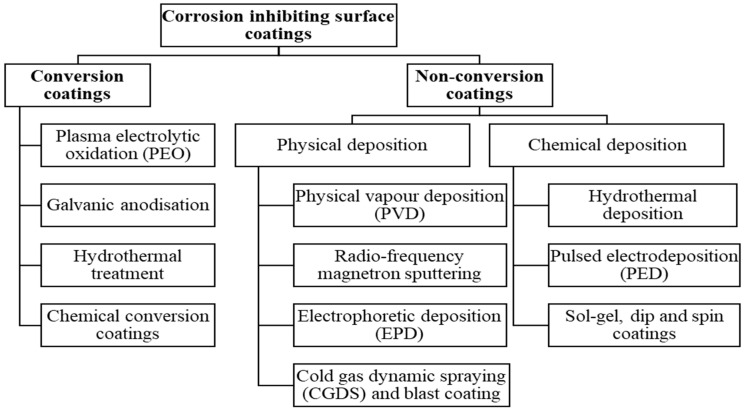
Different corrosion-inhibiting surface coatings for Mg and its alloys.

**Table 2 materials-16-04700-t002:** Distinguishing properties of widely used coating methodologies for Mg/Mg-alloys.

Coating Methodology	Merits	Demerits	Ref
Plasma Electrolytic Oxidation (PEO)/Micro-Arc Oxidation (MAO)	Coatings of different porosity and thickness can be obtained. Improved corrosion, wear and mechanical properties of the substrate can be attained *. Additives/particulates of choice can be added to the electrolyte that integrates into the oxide layer. Improved biomineralization is observed. Simple, in-situ experimental setup.	The porosity of the coatings can cause accelerated corrosion. Post-deposition sealing is required to minimise porosity. Decreased biocompatibility due to the passive surface layer standalone coating.	[21,22,23,24,25]
Anodisation	5–30 µm thick, stable, well-adhered, hard, dense, electrically insulating metal conversion coating. Offers superior corrosion and wear resistance *. High thermal stability, heat and shock resistance, and good optical properties. Less operating voltage (when compared to PEO). Simple experimental setup; meagre initial investment.	Variations in porosity along different zones of the anodised surface highly influence the properties of the anodised layer. Additional sealing and/or painting for ideal protection. Not well established for Mg alloys. Defects caused by metallurgical phases in alloy substrates can cause localised corrosion attacks. High maintenance cost.	[26]
Hydrothermal coating	Simple, efficient, and cost-effective. Uniform, compact, crystalline coating. Enables bioceramics and noble metal incorporation. Offers better corrosion resistance *: considered as a potential alternative to chromium conversion coating.	Observed pitting corrosion in the interlayer between hydroxide and ceramics. Requires elevated temperatures (>90 °C) for a longer time (>1.5 h).	[27,28,29]
Chemical conversion coating	A simple method based on chemical dissolution and precipitation principle, investigation of corrosion resistance *. Low operative cost. Doping of Ca/Ce/Sr/K/F can be achieved by tuning electrolyte composition.	Surface morphology Generally, it reveals a ‘riverbed’ like structure that causes an increased corrosion rate. Established for phosphate-containing electrolytes.	[20,30]
Physical Vapour Deposition (PVD)	Tunable mechanical, aesthetic, and corrosion properties *. Thin-film coatings can be generated.	Only selective coatings can be achieved as the thermal energy input can metallurgically deleteriously alter the substrate instead of improving corrosion and wear properties. Thin coatings: easily susceptible to wear and abrasion losses.	[31,32,33]
Radiofrequency magnetron sputtering (RF-MS)	Ultra-thin coatings uniform (550–750 nm) can be achieved. Improved degradation resistance *.	Metallic phase precipitation on the coating surface can cause localised attacks and coating failure. Specific operating conditions.	[34,35]
Electrophoretic deposition (EPD)	High coating rate with a uniform layer of 0.1–2.0 mm thickness. Improved corrosion resistance *.	Trial-and-error to establish the process parameters and their influence on coating formation. Only applicable for pre-defined electrolyte composition. Coating damage in post-treatment.	[36,37]
Cold Gas Dynamic spraying (CGDS) and blast coating	More or less homogenous composite coating. Superior adhesion of bioceramic coatings is achieved through the plasma spraying technique, which enhances durability. Mechanical surface activation along with simultaneous dense ceramic incorporation. The coating surface offers protection against corrosion and wear attack and inhibits biodegradability *.	High operating conditions (temperature, pressure). Relatively recent and less explored.	[38,39]
Sol-gel	A smooth coating layer of <1 µm can be achieved. Enhanced corrosion resistance.	Specific controlled reaction conditions. Susceptible to cracks in the coating.	[40]
Dip and spin coatings	A uniform coating layer of 0.05–5 μm. Polymer blend coatings of 100 μm thickness are claimed to be achievable. Ease of use. Augmented corrosion rate *.	The requirement of finishing at a high temperature can lead to coating damage.	[41,42]

* Superior resistance than bare alloy and/or pure Mg.

## 4. Inorganic Conversion Coating

Mg with relatively low purity (99.9%) exhibits poor corrosion resistance than ultra-high pure Mg (99.99%) because of the pronounced micro-galvanic effect of the impurities in the former (that are highly cathodic) [43]. Mg alloys suffer pitting as they develop a quasi-protective surface film disrupted by the Cl^−^ ions in the physiological fluid. A robust and continuous hydroxide layer as a coating that may improve corrosion resistance can also enhance cell survival. However, it caused reduced cell densities and abnormal cell morphology, thus making it an inadequate choice as a standalone layer (despite its prospect for improving corrosion resistance) [44]. Effective anodic oxidation of Mg that can be achieved through MAO enhances surface wear resistance because of the ceramic-like oxide film, which is non-compact and porous [45]. Organic polymer coatings and silanes are used as a sealant to plug the surface defects/porosity of MAO that can delay the onset of Mg corrosion [46]. The hydroxyl group at the surface facilitates the bonding of silane with the substrate. Cui et al. [46] observed that the poly methyl tri methoxy silane (PMTMS) coating on MAOised AZ31 did not act as an effective sealant; therefore, they investigated a prior alkali-treatment for developing MAO/PMTMS composite coating.

Strategies to improve bioactivity and corrosion resistance through phosphate-based conversion coatings have been investigated. Surface-activated Mg-Mn-Zn alloys subjected to Ca-P treatment showed improved in-vitro cellular adhesion (Figure 5a) [47]. Immunohistochemistry revealed the coated alloy to have a greater mean optical density (MOD) than the uncoated substrate (Figure 5b). Pathological examination of the bone-implant interface revealed no lymphocytic infiltration or plasmablastic infiltration. Ca-P aids in forming a new bone matrix which is attributed to better extracellular matrix (ECM) activity and new osteoid formation with osteocytes. Early bone growth during the postoperative period is through osteogenesis, and osteoconductivity is achieved using CaP-based coating in the in-vivo model.

Hydroxyapatite (HAp) is a gold-standard material for its chemical similarity with human bone and teeth, bioaffinity and osteoconduction. Mg is essential in DNA stabilization [48], metabolism, and enzyme activation in human systems [49]. The ratio of Ca to Mg affects the formation of HAp as there is continuous replacement of Ca by Mg ions during the alloy’s active dissolution. Inorganic calcium salt precipitation on the Mg substrate, followed by subsequent alkali and heat treatments to form HAp crystals of protective HAp coatings (developed by the sol-gel method), is reported [50]. Chen et al. [51] introduced an electroless two-step route to produce conversion crystalline HAp that improved corrosion resistance of bare Mg in minimum essential medium (MEM), by two orders of magnitude (Figure 6), substantiating the influence of secondary sealant/coating for the primary coating.

Chelated calcium salt that facilitates Ca-P precipitation and decreases the localised Mg ion concentration (Figure 7) improved corrosion protection. However, extended exposure to NaCl led to a decrease in corrosion resistance owing to localised damage in the protective coating [52].

Fluoride conversion coatings on Mg substrate for corrosion protection can cause systemic toxicity in implant application. Although fluorine-based polymer coatings offer wear resistance and improved load-bearing properties, and anti-fouling and anti-bacterial properties, the dissolution of polymer leachate (fluoride ions) can compromise host cytocompatibility that eliminates such coatings as a choice for any biomedical use [53]. Fluoride-doped HAp and its dissolution can improve corrosion resistance and bioactivity, however, any direct and excessive dissolution of fluoride ions into the physiological system poses a biosafety concern [54]. To circumvent this, a surface film over the dopant coating may restrict the ionic dissolution to tolerable limits. Bakhsheshi-Rad et al. [55] have deposited nanostructured fluorine doped-HAp (nFHA)-polycaprolactone (PCL) composite coating with improved adhesion and enhanced nucleation sites for apatite formation. The mixed layer improved corrosion resistance. The polymer degradation mechanism (Figure 8) suggests that the polymeric membrane porosity is the likely site for corrosion initiation, with the underlying nFHA layer [55].

Though the fluorine-free electroless coating improves corrosion resistance, their biocompatibility and bioactivity aspects are not reported. Phosphate-based conversion coatings of Ca, Zn, Mg, Mn, Sr, Ce and K improve the corrosion resistance of Mg alloys, but they may require further modification [30]. Amorphous Ca-P and HAp-based coatings offer good corrosion resistance, bioactivity, and biocompatibility [50]. Several studies employed surface incorporation of bioactive molecules to improve osseointegration and promote bone healing [56,57,58,59]. Besides their porosity, poor toughness may be another limitation with a standalone inorganic/ceramic coating layer that may facilitate wear losses. A flexible coating such as polymeric coatings may be attractive in this respect. 

## 5. Organic Polymer Coatings

For the advantages in the preceding description, biodegradable polymeric coatings are attractive. Further, their tunability to achieve osseointegration and controlled and localised drug delivery give biodegradable polymer coatings an edge over other coatings for Mg alloys in implant application [60]. However, such coatings must be biocompatible for such applications [61], besides the primary needs of surface coverage and stability until the healing has been achieved [62]. Polymer molecular weight and the number of coating layers influence corrosion resistance and the concurrent hydrogen evolution (Equation (1)). Biodegradation of the polymer coatings occurs by water adsorption and hydrolysis, and these degradation products may either be eliminated through body fluid or incorporated into metabolic pathways [63]. Polymer coatings can also be functionalized for bioactivity, drug delivery, self-healing, and osteo-inductance, making them a preferred choice over other surface coatings [36]. Figure 9 presents structure of some of the commonly employed synthetic and natural polymer-based coating systems. 

### 5.1. Natural Polymeric Coatings

Natural polymers such as chitosan, collagen, and gelatin that are advantageous in achieving better cell attachment, growth, and tissue-implant integration are commonly introduced onto Mg alloy surface through dipping, spin-coating and electrophoretic deposition [62]. Natural polymers are mostly polysaccharides and proteins that exhibit biocompatibility and biodegradability. Cationic chitosan and anionic alginate, polysaccharide cellulose and commonly sourced physiologically natural proteins such as collagen and albumin are used as a sealant to the pretreated Mg alloy for various reasons as described in Table 3.

### 5.2. Synthetic Polymeric Coatings

Synthetic polymers are largely preferred over natural polymers for the predictability/consistency of their properties (such as corrosion protection, mechanical properties, etc. [62]). Some commonly studied synthetic polymer coatings on Mg alloys, described in Table 4, are discussed subsequently.

(a) Porous polycaprolactone (PCL) membranes were spray coated on AZ91 ingots to control the degradation under in-vitro and in-vivo conditions [58]. Enhanced Green Fluorescent Protein Osteoblasts (eGFPOB) were seeded onto the coated and uncoated substrates under standard culturing conditions to evaluate biocompatibility. A micro- or nanoporous structure mimicking the extracellular matrix surface facilitates faster cell adhesion and growth. The coated alloy maintained its compressive strength during 60 days of immersion, whereas the strength of the uncoated alloy deteriorated. Porosity-modulated PCL membrane-coated Mg implants inserted into rabbits’ decorticated sites developed no gas pockets, indicating that the circulation was efficient in buffering and nullifying the effects of hydrogen formation. However, the hydrogen evolution developing into bubbles is profoundly governed by the choice of in-vivo system, implantation site and implant size. It is important to note that an accelerated hydrogen evolution in the initial corrosion stages can gradually subside with the substrate’s evolving surface passivation [96]. MTT (3-[4,5-dimethylthiazol-2-yl]-2,5 diphenyl tetrazolium bromide) assay suggested that Mg affects viability with a localised increase in Mg ions at levels exceeding 150 ppm. On the other hand, histological observation showed no harmful effects in the uncoated alloy, even though the Mg ion release was greater due to homeostasis regulation. PCL coating significantly decreased the Mg ion release initially, and even after coating degradation, the effects due to the excessive local release of Mg ions were not alarming [58]. Electrospun nanofibers of PCL deposited on AZ91 showed increased hydrophobicity that delayed the onset of Mg corrosion in SBF (though little information was provided on the degradation behaviour and cytocompatibility) [97]. Further investigations are required.

(b) Poly (trimethylene carbonate) (PTMC), a surface-eroding polymer for Mg alloys, has been evaluated for potential cardiovascular stent application [98]. PTMC surface deposited (by solvent evaporation) onto a high purity Mg alloy maintained ~55% of its original thickness during 16 weeks of subcutaneous implantation in Sprague Dawley rat models. PTMC provided superior corrosion resistance compared to the bare and PCL-coated substrates. Dynamic degradation tests in the modified SBF at 37 °C and physiological pH exhibited significantly lesser Mg ion release and pH change from the surface than with PCL or the bare substrate (Figure 10i). Post-corrosion morphology of the PTMC-deposited surface was homogenously flat, while PCL caused non-uniform bulk erosion, suggesting PTMC to be more promising than other organic coatings (Figure 10ii) [99].

Bare and PTMC-coated wires of Mg-Zn-Mn alloy were implanted in Sprague Dawley rat models, and the coated implantation site showed excellent tissue response [81]. Polycarbonates, polyanhydrides, and polyorthoesters-based surface-eroding polymers show a slower degradation rate than the bulk-eroding PCL-like polymers. PTMC degradation follows almost a uniform surface erosion mechanism [85]. However, it does not remain defect-free to control the underlying substrate and electrolyte interaction. PTMC coating showed decrease in release of Mg ions than the bare and PCL coating. Water diffusion through the PTMC layer was limited. Considering the mobility of the H and OH ions through PTMC coating and interfacial hydroxide formation after an electrochemical reaction with Mg, no visible swelling and an intact coating were seen. It has been proposed that the hydroxide and thin layer PTMC form an enhanced corrosion barrier. In contrast, the PCL-coated substrate showed cathodic hydrogen evolution and increased corrosion rate with Mg corrosion products at the interface. Activated platelets and adherent erythrocytes were aggregated in PCL-coated substrates, whereas the PTMC surface did not show aggregation implying intact fibrinogen [99].

PLGA is a synthetic co-polymer that can be engineered to the desired physical, mechanical, chemical and degradative properties by tailoring the ratio of the two co-monomers. Lactic and glycolic acid degradation products integrate with the normal metabolic pathway, ensuring non-toxic elimination into the physiological system (Figure 11i). Weight loss probed through immersion studies revealed that the coated alloy initially exhibits a lag in weight loss due to the barrier effect of the surface polymer coating (Figure 11(iia)). However, as the medium permeation through the pores and defects increased, the coated alloy followed a similar trend over prolonged exposure to the aqueous solution (Figure 11(iib). This behavior may make PLGA coating promising for its ability to delay degradation, which is one of the main objectives of applying coatings on Mg alloys for temporary bioimplants. However, Li et al. [89] established that increasing PLGA coating thickness does not improve corrosion resistance. For example, corrosion resistance data extracted from PDP curves, as shown in Figure 12, show similar corrosion resistance due to coatings with two different contents of PLGA (that produced different thicknesses). However, SEM revealed that the coating with 2% PLGA showed greater resistance to breakdown. Biocompatibility and cell adhesion make PLGA suitable for biomedical applications. The inconclusive findings seen in Figure 11 [100] and Figure 12 [89] suggest the need to establish a definite role of the synthetic co-polymer PLGA in providing corrosion resistance to Mg alloys. 

Polyhydroxyalkanoates (PHA) is a hydrophobic biocompatible polymer employed in drug delivery systems, tissue engineering and skin replacement [101]. Polyhydroxy butyrate (PHB), poly-hydroxybutyrate-co-hydroxyvalerate (PHBV), poly-hydroxybutyrate-co-hydroxyhexanoate (PHBHx) and polyhydroxyoctonate (PHO) have been extensively studied as a biomaterial for their established biocompatibility and prolonged intactness. PHB possesses superior mechanical and biological properties and ease of fabrication. In addition to these properties, PHBV showed low immunogenicity and favourable fibroblast cell growth. These synthetic polymers may be investigated for their application on Mg alloys for temporary implant applications. The unexplored aspects of the degradation of such polymers (e.g., PHBV [102] include the understanding of the interplay of time, temperature, conditions of the degradation, and their chemical composition and physical states (such as granule size, melt cast film, and solvent cast film).

Polymer molecular weight and the number of coating layers influence corrosion resistance and the concurrent hydrogen evolution [103,104,105]. It is necessary to reiterate that standalone polymer coatings, especially polyelectrolyte coatings of polysaccharides, do not provide the required degree of resistance to water and ionic diffusion toward the substrate due to their inherent porosity. Hence, composite coatings with impregnated bioactive compounds need to be explored to improve the corrosion resistance of Mg alloy implants. 

## 6. Advances in Polymer Coatings for Mg Alloy

Biodegradable polymeric materials are attractive due to advantageous functionalities such as anti-microbial activity, targeted drug loading and release, self-healing, required biocompatibility and enhanced corrosion resistance. Li et al. [106] reviewed the most widely studied synthetic polymers, such as PLS, PLGA, PCL, PDA, natural chitosan, and collagen-based polymers, for their corrosion resistance and biocompatibility. The degradation mechanism and the decline in the mechanical strength of the standalone surface polymer coatings have not been studied in detail for the established natural and synthetic polymer systems. Defect-free surface polymer coatings are ideal, and hence, polymer systems are used as a sealant to the underlying Mg alloy conversion coatings for improved coating adhesion and corrosion resistance [45,107,108]. Bioactive molecules loaded with polymer coatings to achieve improved functionality are discussed in Section 6.3. Moreover, exploring antibiotic-infused coatings among drug-eluting coatings as drug-releasing implants (DRIs) for localised drug delivery is also briefly discussed in Section 6.1 and Section 6.2. This section discusses the established functionalities achieved in surface polymer coatings for Mg substrates and other implant materials that can be extended to the Mg alloys.

### 6.1. Anti-Microbial Coating

Biodegradable polymeric scaffolds imbibed with anti-microbial agents have been extensively reported in drug delivery and surgical sutures [109]. A multifunctional double-layer spray-coated silane onto the traditional SS implant followed by electrophoretic deposition of biopolymers with silica gentamicin nanoparticles was adopted. The composite coating exhibited good adherence and mechanical properties that should be further evaluated for the corrosion performance and drug release profile [110]. Extension of these composite coatings for temporary implant biomaterial can be employed to achieve versatile applications. Further antimicrobial blends based on lignin, chitosan and cellulose-based polymer are being investigated [111]. Polymer composites of PHA/PVA membranes loaded with levofloxacin allow tailoring of the dosage to achieve a localised anti-bacterial effect that can be extended as the physisorbed coatings on Mg substrates [112]. Similarly, PCL-based multi-layer coating with Levofloxacin and HAp on AZ31 showed controlled wettability, degradation, and drug release kinetics [113]. Introducing alloying elements such as Cu in Mg has shown to decrease *Candida albicans* viability. However, the galvanic coupling between Mg and Cu has demonstrated an increased corrosion rate [114]. Future studies to optimise alloying content or doping of Cu in Mg alloy can improve corrosion resistance and antimicrobial activity.

### 6.2. Drug-Eluting Coating

Localised drug delivery using DRI has been extensively studied in temporary and permanent implants to deliver anti-microbial agents and bone growth factors [115]. Polymer coatings control the targeted drug delivery and can be extended to the surface modifications for corrosion control of Mg alloys [116]. Asymmetric poly ether imide (PEI)/sirolimus-loaded PLGA coating on WE43 with base PEI that offers stability to the drug-loaded PLGA reduces the initial sirolimus burst release (inset Figure 13A) as well as acts as a corrosion barrier. The underlying PEI membrane does not participate in the drug loading and release (Figure 13B) [117].

### 6.3. Bioactive Coatings

Adding bioactive compounds such as Ca-P and HAp, i.e., osteoconduction enhancing compounds, to the polymeric coatings has improved corrosion resistance more than the bare substrate [118,119]. Bioactive ceramics are brittle and have mechanical properties different from human cortical bone, causing stress shielding effect and implant loosening. Hence, in a combinatorial approach of reducing porosity in the polymer coating and increasing bioactivity, molecules for mimicking extracellular matrix have been employed in the composite layers. PLGA-hardystonite composite coating on sintered Mg alloy decreased the corrosion rate in Kokubo SBF and increased the Ca-to-P ratio, indicating improved bioactivity. In addition, the organic-ceramic layer also expedited HAp formation during immersion in SBF (c.f., the organic coating alone). However, the dissolution of the corrosion product and the bioactive ceramic eventually exposes the underlying PLGA layer. The polymer decomposes to form a negative surface charge that attracts positive ions such as Zn and Ca from the physiological system along with surface passivation of the underlying Mg substrate, cumulatively accounting for the prolonged surface bioactivity aiding bone healing [91].

### 6.4. Self-Healing Coatings

Developing a corrosion product layer on the Mg surface protects the underlying substrate [55,108]. Scratch test on composite MAO/PTMS coated AZ91 revealed sealing (self-healing) of the scratch during 24 h immersion in a chloride environment. However, a high volume of the hydroxide layer (than Mg) and localised hydrogen bubble formation can induce localised stress, causing delamination of PTMS in prolonged exposure [46]. Self-healing coatings for high electrochemically active Mg-based alloys may be an ineffective approach as it could lead to accelerated cathodic activity.

### 6.5. Self-Assembled Coating

Advances in polymer chemistry have shown considerable interest in self-assembled monolayers (SAM) for their intrinsically simple structure at their interfacial region at the surface functionalized polymer. Exploiting the chemisorption of the thin organic layer, Qiao et al. [120] reported long-chain fatty acids comprising hydrophobic long-chain hydrocarbon, and the hydrophilic carboxyl group are chemically adsorbed on the cationic Mg substrate. SAM is mainly of magnesium hydroxide, magnesium oxide and other magnesium compounds containing hydrogen and carbon that develop upon interaction with the environment. Carboxylic carbon was not detected in organic polymers such as oleic acid and stearic acid in various solvents, indicating the absence of unreacted fatty acids. Anodic polarization of the SAM-coated AZ31 showed a nobler shift in the pitting potential than the bare alloy. As the overall corrosion rate is not significantly decreased, the authors recommend the thin SAM film as a post-treatment strategy [121]. 

### 6.6. Layer-by-Layer Coating

Composite coatings have been discussed as combinatorial and non-conversion coatings or physisorption of organic and ceramic biomaterials for improved functionality [106]. A pre-existing Mg(OH)_2_ layer improves the bonding of the substrate with the organic coating having suitable functionality. However, the quasi-protective nature of the corrosion product layer and the accumulation of interfacial Mg corrosion products can cause delamination of the surface polymer film. Stearic acid (SA), a biocompatible molecule that can tether the hydroxide and PTMC film, decreases Mg corrosion and hydrogen evolution. The synergistic effect of the hydroxide, SA and PTMC leads to decreased permeation of the PBS and improved corrosion resistance compared to other coatings and bare AZ31 in up to 30 days of immersion (Figure 14a–c). In-vivo evaluation in rat femur revealed the lowest volume loss in Mg-OH&SA-PTMC compared to other coated substrates at the end of twelve weeks (Figure 14d). In contrast, single-layer Mg-OH and composite Mg-OH&SA showed superior corrosion resistance than the bare AZ31 for 12 weeks, but accelerated corrosion afterwards. The sandwiched coating of Mg-OH&SA-PTMC also exhibited improved cellular adhesion and bone formation in rats, which was attributed to the gradual corrosion acceleration of the coated implants (Figure 14e) [122].

## 7. Conclusions

Metallic materials have been the preferred class of biomaterials for orthopaedic applications for their load-bearing properties. Mg-based biodegradable implants offer biocompatibility, bioresorption and easy elimination of the biodegradation products from the host system, making them a promising candidate for temporary implant applications. High electrochemical and corrosion activity deters the immediate use of Mg-based implants, necessitating surface modification. An ideal surface coating for temporary implant application is expected to fulfill the requirements of biocompatibility, biodegradation, and non-toxic degradation products in-vivo with easy elimination from the physiological system without causing biosafety concerns. Commonly studied techniques to develop conversion and non-conversion coatings on Mg substrates for improved corrosion resistance have been discussed with their merits and limitations. The discussion includes the conversion of inorganic coatings and natural organic polymer coatings, their sources and key features, common synthetic polymer coatings, their biocompatibility, surface coating methodologies, and their merits and limitations. PCL and PTMC-based polymers with in-vivo evaluation and degradation mechanisms of common polymers are also discussed. Among the different coating approaches investigated, no single coating recipe addresses degradation control and functionality. Controlled degradation of the surface film with improved functionality to delay the onset of Mg corrosion is the key to their translational evaluation. Based on the advantages and limitations of different coating types, this review argues the greater prospects of composite coatings.

## Figures and Tables

**Figure 1 materials-16-04700-f001:**
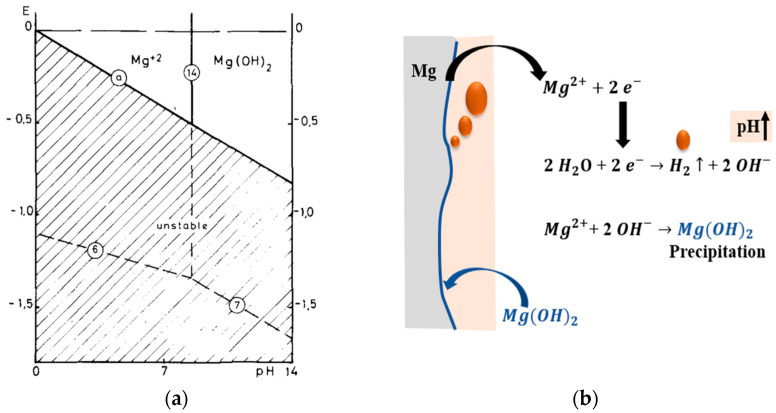
(**a**) Pourbaix diagram [10] of Mg-H_2_O system (25 °C and 1 atm) (reproduced with permission from Elsevier), and (**b**) schematic representation of chemical and electrochemical reactions at the surface of Mg when exposed to an aqueous environment (redrawn from [11]).

**Figure 2 materials-16-04700-f002:**
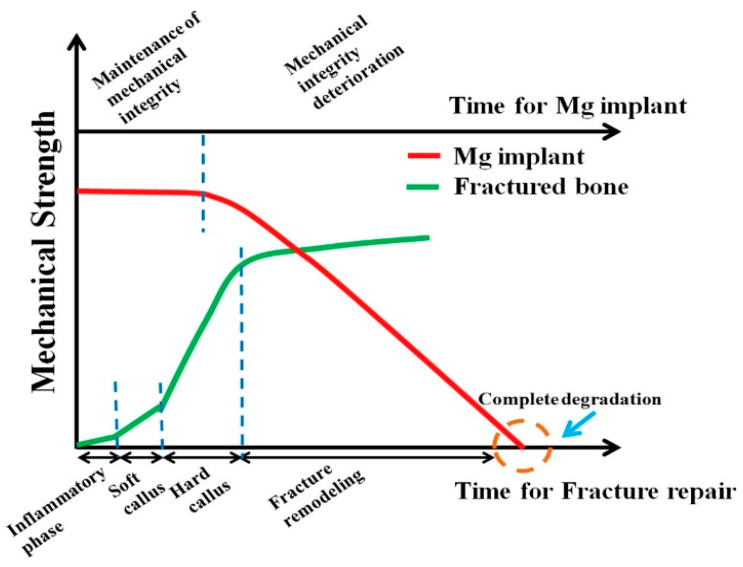
Ideal degradation behaviour of Mg-alloy biomaterials in bone-fracture healing [14] (reproduced with permission from Elsevier).

**Figure 5 materials-16-04700-f005:**
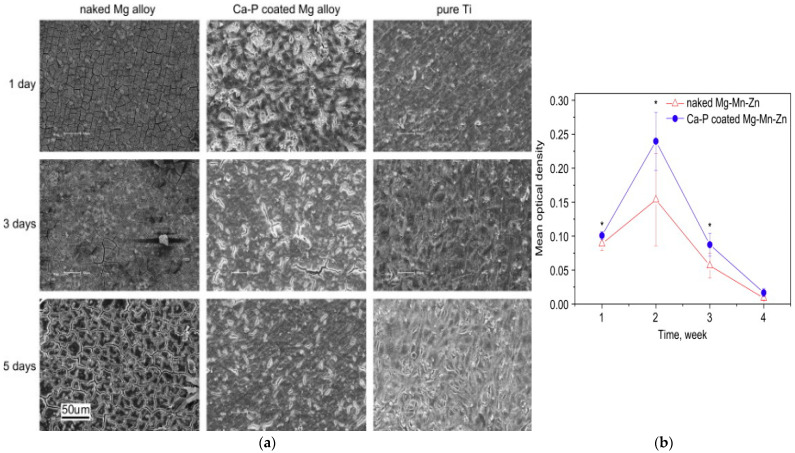
(**a**) Cell morphology after incubating for 1, 3 and 5 days on different materials. (**b**) MOD values of TGF-β1 expression at the interface between the implant and the bone after different implantation periods [47] (reproduced with permission from Elsevier).

**Figure 6 materials-16-04700-f006:**
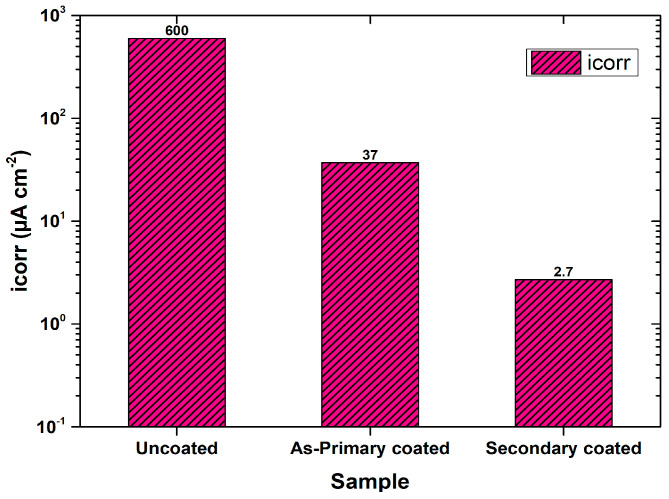
Corrosion current of MEM for uncoated high purity Mg, as-primary coated, and secondary-coated (HAp) Mg (obtained from potentiodynamic polarisation (PDP) plots in [51]).

**Figure 7 materials-16-04700-f007:**
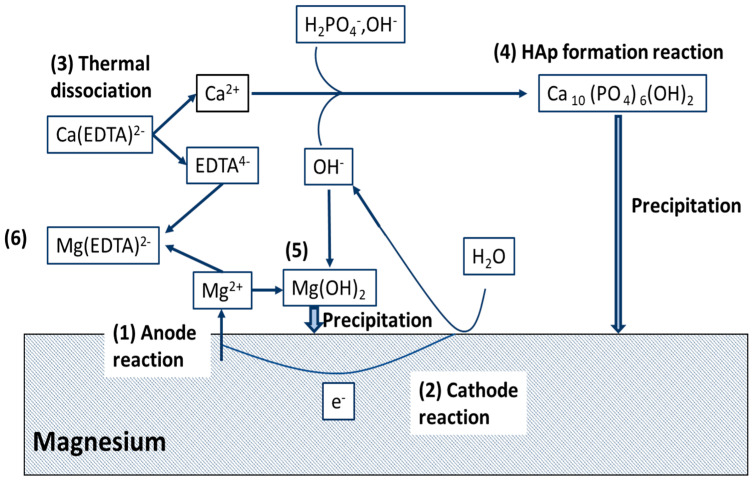
Schematic illustration of the precipitation mechanism of HAp on magnesium in the Ca-EDTA aqueous solution [52] (reproduced with permission from Elsevier).

**Figure 8 materials-16-04700-f008:**
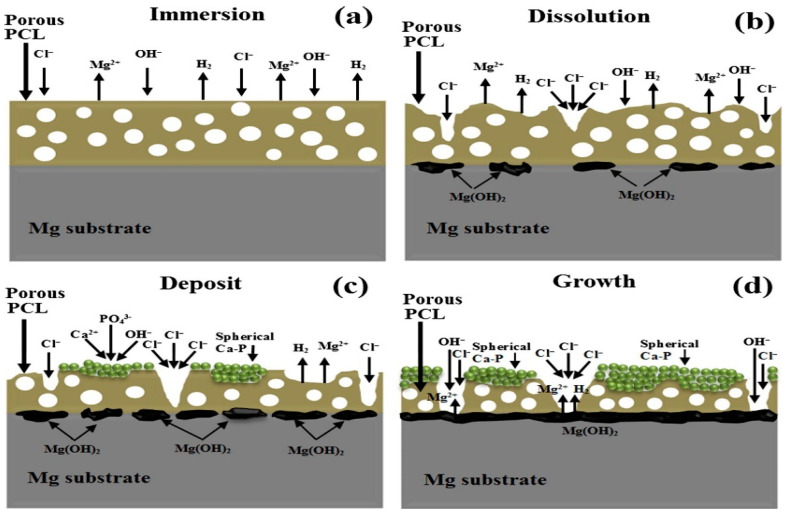
Schematic illustration of mechanism of: (**a**–**d**) progress of the degradation of the porous PCL layer after immersion in Kokubo simulated body fluid (SBF) solution [55] (reproduced with permission from Elsevier).

**Figure 9 materials-16-04700-f009:**
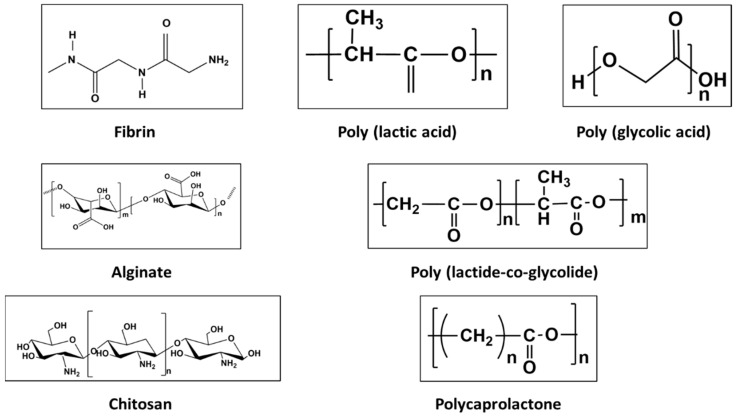
Chemical structure of commonly employed natural and synthetic polymers [64,65] (Reproduced with permission from Elsevier).

**Figure 10 materials-16-04700-f010:**
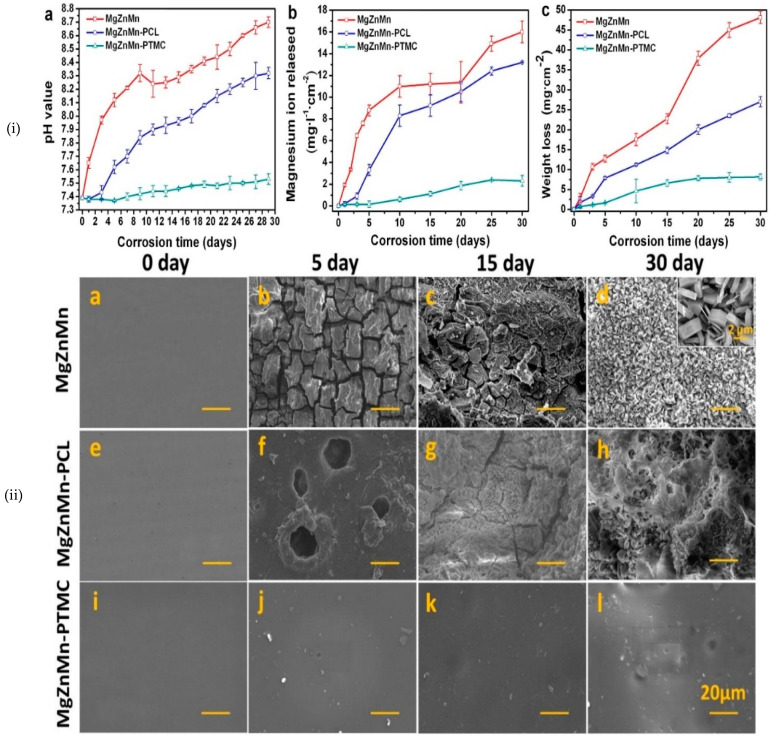
(**i**) Measure of (**a**) change in pH; (**b**) concentration of magnesium ion release; (**c**) weight loss of the specimens with time during the dynamic degradation of the uncoated, PCL-coated and PTMC-coated Mg-Zn-Mn alloy in m-SBF. (**ii**) Scanning electron micrographs of the surface morphologies of: (**a**–**d**) the uncoated, (**e**–**h**) PCL-coated and (**i**–**l**) PTMC-coated Mg-Zn-Mn alloy after immersion in m-SBF for different durations (0, 5, 15 and 30 days) [99] (reproduced with permission from Elsevier).

**Figure 11 materials-16-04700-f011:**
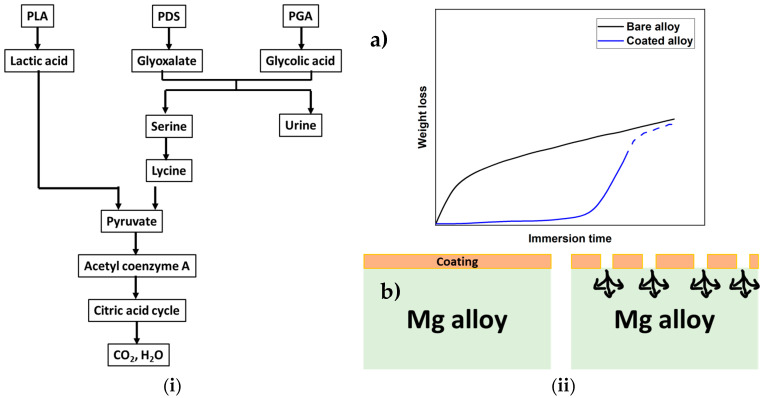
(**i**) Degradation pathway of the common synthetic polymers (Redrawn from [100]); (**ii**) Proposed degradation: (**a**) kinetics and (**b**) mechanism for PLGA-coated and bare Mg alloy (Redrawn from [89]).

**Figure 12 materials-16-04700-f012:**
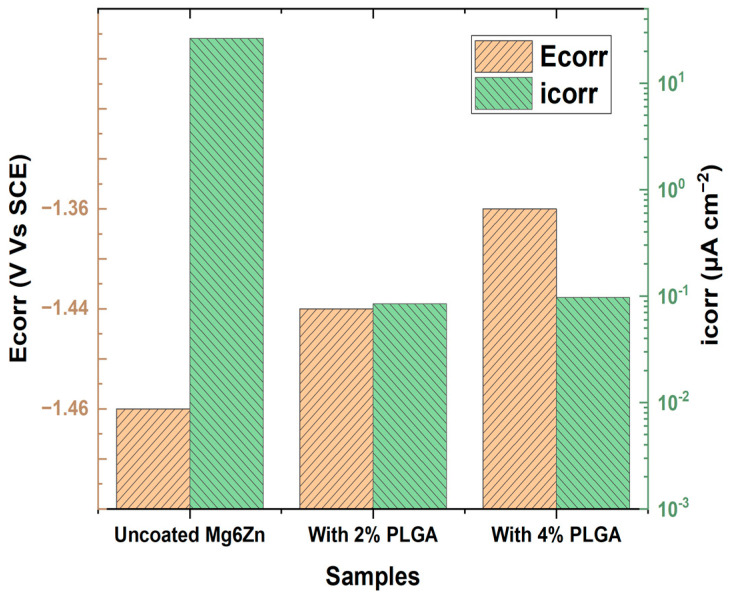
Electrochemical corrosion potential and current density obtained from potentiodynamic polarisation (PDP) curves for bare, and PLGA coated Mg alloy in 0.9 wt% NaCl [89].

**Figure 13 materials-16-04700-f013:**
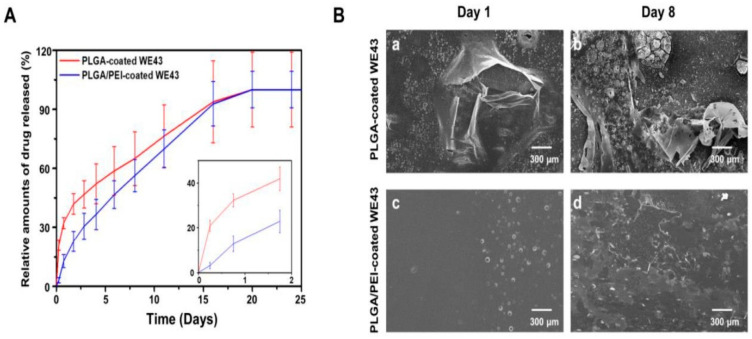
(**A**) Drug release profiles for PLGA, PLGA/PEI coated WE43 in phosphate buffer saline (PBS); (**B**) representative scanning electron microscopy images for (**a**,**b**) PLGA-coated WE43 alloy, and (**c**,**d**) PLGA/PEI-coated WE43 alloy after 1 and 8 days of immersion [117] (reproduced with permission from John Wiley and Sons).

**Figure 14 materials-16-04700-f014:**
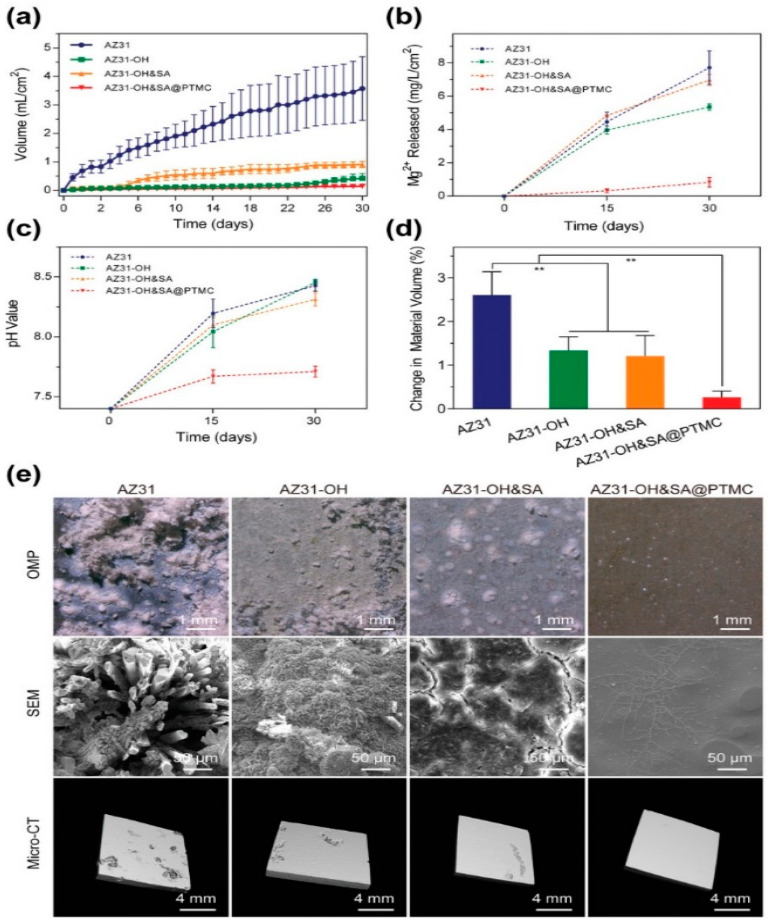
In-vitro degradation behaviour of AZ31, AZ31-OH, AZ31-OH&SA, and AZ31-OH&SA@PTMC as a measure of (**a**) released H_2_ volume, (**b**) Mg^2+^ concentration released, (**c**) pH value, and (**d**) change in material volume for bare and coated AZ31 for immersion degradation in PBS solution at 37 ± 0.5 °C up to 30 days (** *p* < 0.01), and (**e**) optical microscopy (OMP), scanning electron microscopy (SEM) and micro-computed tomography (micro-CT) [122] (reproduced with permission from John Wiley and Sons).

**Table 1 materials-16-04700-t001:** Comparison of mechanical properties of Zn, Fe, Mg, and the natural bone [2].

Property	Zinc	Iron	Magnesium	Natural Bone
Density (g/cm^3^)	7.13	7.874	1.74–2.0	1.8–2.1
Elastic modulus (GPa)	90–110	180–210	41–45	3–20
Yield strength (MPa)	75–160	50–1450	65–100	130–180
Fracture toughness (MPa m^½^)	35–120	120–150	15–40	3–6

**Table 3 materials-16-04700-t003:** Commonly employed natural polymers as coatings for Mg corrosion control.

Polymer	Source	Composition	Key Features	Ref.
Chitosan	Exoskeleton of crustaceans and insects such as butterflies, ladybugs, lobster, coral, crab	Cationic polysaccharides made of N-acetylglucosamine and D-glucosamine units.	Ability to chelate metal ions. Promotes cell attachment.	[66,67,68]
Alginate	Brown algae	Anionic polysaccharides comprise 2 monomers (1,4) β-D-annuronate (M) and α-L-guluronate (G).	Tailor G-group for improving mechanical properties and M-groups for modulating immunogenicity. The addition of other bioactive compounds (e.g., fibronectin) is required to facilitate cell growth.	[69,70,71]
Cellulose	The structural component of plants	β-linked D-glucose units	Non-immunogenic. Good processability. Scarce data as a coating for corrosion resistance of Mg alloys.	[72]
Collagen	Three types are based on the tissue present. Extracellular matrix (ECM) of bone (Type I); cartilage (Type II); blood vessel wall (Type III)	Triple helical polypeptide domains	Non-immunogenic; antigenic response; osteoblast adhesion and migration. Aids in osteoinduction.	[73]
Gelatin	Denatured collagen	Polypeptide	High solubility. Low polymer production cost.	[67,74]
Albumin and fibrin	Human blood plasma protein.	Albumin-small globular protein arranged in 3 repetitive homolog domains, Fibrin-formed from fibrinogen.	Abundant protein in human plasma facilitates protein adsorption on the surface. The homogenous, high-purity coating is a challenge.	[75,76,77]
Octadecanoic acid	Glycerol ester in animal and plant fats.	A saturated fatty acid with long chain CH_3_(CH_2_)_16_COOH	Polar head groups can bind with metal cations.	[78]
Phytic acid	Edible nuts, cereals and legumes.	Myo-inositol hexaphosphoric acid (C_6_H_18_O_24_P_6_)	The ability of the active group to bond with metal ions to form chemical conversion coatings for corrosion resistance remains under-explored. Phytic acid-based conversion coatings showed good adherence to the Mg substrate. Composite coating to facilitate osseointegration also improved corrosion resistance.	[79,80,81]

**Table 4 materials-16-04700-t004:** Some of the widely studied synthetic polymer coatings on Mg alloys.

Polymer	Advantages	Disadvantages	Remarks	Ref.
PLA/PLLA (Poly L-lactic acid)	Low non-toxic biodegradation. Degradation by hydrolysis.	Not suitable for load-bearing applications. Erosion through random hydrolysis	Spin coating methodology can be employed. The adhesion of the polymer alone coating is very poor. Surface modification of metals/ ceramic-polymer composites was suggested.	[82,83,84,85]
PLGA (poly lactide-co-glycolic acid)	Biodegradation; non-toxic by-products. (approved for human clinical trials) Tunable degradation properties by modifying the ratio of the two monomers.	Cathodic delamination and surface layer bulging in short immersion time.	Dip coating methodology is widely reported. Surface pre-treatment by MAO improves SCC resistance. Ductile PLGA for stressed implants.	[86,87,88,89]
PCL(Poly-caprolactone)	US-FDA approved. Low degradation rate and biodegradability; controls alkalinisation. Prevents galvanic cell formation in bare/plasma bonded substrate, thereby augmenting corrosion resistance. High strain failure rate, ease of processing (can be used along with organic solvents, wide temperature range of operation: −60 °C to 60 °C)	Poor adhesion.	Dip coating. Composite coating methodologies such as plasma treatment and surface alkalinisation to further improve polymer adhesion to substrate and corrosion resistance.	[41,42,55,90,91,92,93,94,95]

## Data Availability

No new data included in the review article.

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
