# Peer review of "Polymeric Coatings for Magnesium Alloys for Biodegradable Implant Application: A Review"

_materials, 2023, doi:10.3390/ma16134700_

Round 1

Reviewer 1 Report

The topic addressed in the manuscript is very important for the field. However, the paper is not well organized to adequately convey this valuable information.

The writing in sections after the Introduction needs to be improved, because the text is not fluent and easy to follow. Reader has a feeling that the reviews on the papers are just enumerated one after another. In every section of this kind of a paper authors should give some general, info and prospective on the topic, explain the characteristics of existing groups of materials/treatments and then perform a review of a group. In this review manuscript in a lot of cases the reader does not get the info why the reviewed (chosen) articles are so important for the fields.  Most of the topics (types of coatings) are covered only with one reference, it seems to reader that some of these materials are isolated cases. All this has to be changed throughout the manuscript

In the introduction section, in order to better convey the importance of this topic, explain in greater details  the stress shielding effect.

Authors should give some explanations on the general strategy when biodegradable coatings are applied to Mg implants. Different approaches that are applied for control of Mg alloy degradability should de addressed in more detail. In a lot of places the comparison between different materials is missing. All this has to be changed throughout the manuscript

Row 108. You should add few sentences why you chose to focus on biodegradable surface coatings.

Table 2.is not introduced in text near the table, nor explained its content. Authors should do so.

In Table 2. The statement “Non-uniform deposition and metallic phase precipitation on the coating surface can cause coating failure” I do not agree with this statement, PVD coatings are uniform but they are not easily deposited inside deep holes, and geometries with narrow trenches or so. Therefore, you should modify this statement, or more properly explain.

Table2. Should contain information about the types of the coatings that can be deposited by specific method. Because, not all mentioned techniques can produce all coatings.

Rows 117-124, 129-131, 146-150,167-170,183-184, 233-238, 249-263,  310-311, 334-336, 349-351,  – Information in these sentences is taken from some literature/works, therefore in these cases adequate citation is required.

Figures 5, 6, 10, 14 – the resolution/quality of the images, diagrams and text on listed figures should be improved.

The review paper should be systematic, well organized and easy to follow. The parts of the work where this is not the case are :

·         - First paragraph of Section 4., should give an introductive perspective on the topic that is covered in that section. On the contrary it contains some information not connected to the text around in manuscript.

·         - Section 4.2.2., the reviews on the papers are just enumerated, and written just one after another without interconnection between them. This section also does not have an introductive part so the reader does not have an idea what is all about.

·         - Row 289-this part is not logically and fluently connected with other parts of the manuscript. Rewrite and reorganize these parts

·         - Section 5.1., just enumeration of reviews one after another, without interconnection with the rest of the paper.

·         - Section 5.4. and start of the section 5.5. interconnection with the rest of the paper is not clear, just enumeration of the reviews… give an introductive perspective on the topic, otherwise remove this subsection .Because, the question is that are these subsections justified?

 Section 4.1. , 1st paragraph. Is this the only work that addressed that addressed this treatment? If not then you should mention, cite more works. Given that this is a review paper this should be done throughout the work.

Figure 6. should be briefly addressed in the manuscript.

Section 4.2.1- Basically the whole paragraph is based on the information given in Table 4. Authors should give some more general information about the natural polymeric coating properties, their behavior, usage and applications.  This paragraph seems unfinished.   

Section 4.2.2. authors should give some general information on these materials, kinds, and then perform  a review.

“Table 5. Some of the widely studied synthetic polymer coatings on Mg alloys” – This should be introduced on the start of the section, and as said in previous comments these groups of materials should be addressed in the text.

Sentences in rows 265-268 provide just some brief information about this material, and a big figure 11 which is not justified by text. This has to be improved, and this part of the manuscript should be connected with the rest, and some explanations about Figure 11 should be given.

Sentence in Row 272-  Specify which coatings?

Ambiguous sentences which should be rewritten: Rows 294.

Section 5.1. lacks interconnection between the sentences, the thoughts in it are not fluent an it is not easy to follow

Section 5.2.  gives just very brief and partial information about the mentioned material. This has to be improved. It seems that these sentences along with Figure 13 are just given to fill the space. This part should be elaborated in more detail and explained, or removed. The same is for some other parts with similar issues.

Figure 14 is quite complex but its resolution is not satisfactory, some parts of it are not really needed. This image requires explanation in the text too.  Lowest part of Figure 14.e seems not really needed either remove or should be explained.

Most of the conclusions do not address the topics reviewed by this paper. Therefore, these should be brought in line with the findings obtained in this review.

Author Response

Please see the attached doc.

Reviewer 2 Report

Mg-based biodegradable implants offer various advantages. This study examines the benefits and drawbacks of several existing coating methods, which can be generally classed as conversion and non-conversion coatings. Still, there are some modifications that I suggest for this manuscript before being accepted for publication.

1. Abstract is still ambiguous, it should highlight the content of the manuscript, at first read of this abstract one can not understand the clear topic of this review, so please re-write the abstract. You should extend the explanations in this sentence: " Coatings are amongst the common approaches to address this challenge. " The need for coatings in Mg and its alloys should also be briefly mentioned at the beginning of the abstract to justify the utility of this review.

2. Some images in the manuscript have a poor resolution, this is unacceptable. Please find a manner to provide high-quality images or re-draw them. I.e. Fig 6 and Fig 7 one can see the pixels of the image, also valid for other images (Fig.9, Fig.14, etc.).  For instance, the chemical structures in Figure 9 can be easily redrawn in ChemDraw or similar software.

3. Do you have permission to republish the images and Tables that were used in your review? I could not see this mentioned in this review paper

I recommend a major revision of this manuscript since it was very difficult to follow the information due to the poor quality of the images. 

Moderate editing of English language required

Author Response

Please see the attached doc.

Reviewer 3 Report

The authors studied Polymeric Coatings for Magnesium Alloys for Biodegradable Implant Application

 There are several problems to be addressed:

1.        The authors should provide Figure 6 in higher quality.

2.              Table 3 is missing. Renumber the tables in the correct order.

3.      The authors should format the article according to the journal requirements. For example, references to the article should be given in square brackets "[12]" and in the order in which the article is mentioned.

4.             Please add more references related to this theme

5.            I suggest improve the conclusion section. The authors should outline the most appropriate coatings for improve magnesium corrosion properties. The conclusion about composite coating should be more specific based on literature.

Minor editing of English language required

Author Response

Please see the attached doc.

Round 2

Reviewer 1 Report

Dear authors, by answering to the previous comments and remarks you significantly improved the manuscript. However, I still have some remarks which should be checked by you:

Row 45 – It`s not the “stronger material” it is a more rigid, or material with higher modulus of elasticity. This sentence does not explain sufficiently the stress shielding and the consequences that it induces. Therefore, this sentence have to be rewritten and expanded with one or two additional sentences.

Figure 3- Quality of figure should be improved.

Ro 140, - Ambiguous sentence, have to be rewritten use more commas to separate parts of long sentences.

In a previous review it was suggested that Table 2. should contain information about the types of the coatings that can be deposited by specific method. Because not all mentioned techniques can produce all coatings.” Authors avoided adequately responding to this remark, therefore once again I request to include such information to make the article more comprehensive.

Paragraphs starting in rows 244 and 267 – It is not clear to what does stands (a) and (b) and as already said in the previous review, these parts lack connection with the previous text. It seems that these sentences start just out of the blue. Adopt the start of these paragraphs to make them more fluent.

The language is mostly good, but there are really a lot of long sentences which should be split in two...

Author Response

Please see the attached doc.

Reviewer 2 Report

The authors improved the manuscript according to the suggestions received. I recommend its publication in the present form.

Minor editing

Author Response

Please see the attached doc.